



# Testing the Reliability of Interpretable Neural Networks in Geoscience Using the Madden-Julian Oscillation

Benjamin A. Toms [1], Karthik Kashinath [2], Prabhat [2], and Da Yang [2,3]

[1]Department of Atmospheric Science, Colorado State University, Fort Collins, CO
[2]Lawrence Berkeley National Laboratory, Berkeley, California
[3]University of California, Davis, Davis, California

**Correspondence:** Benjamin Toms (ben.toms@colostate.edu)

**Abstract.** We test the reliability of two neural network interpretation techniques, backward optimization and layerwise relevance propagation, within geoscientific applications by applying them to a commonly studied geophysical phenomenon, the Madden-Julian Oscillation. The Madden-Julian Oscillation is a multi-scale pattern within the tropical atmosphere that has been extensively studied over the past decades, which makes it an ideal test case to ensure the interpretability methods can recover

the current state of knowledge regarding its spatial structure. The neural networks can, indeed, reproduce the current state of knowledge and can also provide new insights into the seasonality of the Madden-Julian Oscillation and its relationships with atmospheric state variables.

The neural network identifies the phase of the Madden-Julian Oscillation twice as accurately as linear regression, which means that nonlinearities used by the neural network are important to the structure of the Madden-Julian Oscillation. Interpre-

tations of the neural network show that it accurately captures the spatial structures of the Madden-Julian Oscillation, suggest that the nonlinearities of the Madden-Julian Oscillation are manifested through the uniqueness of each event, and offer physically meaningful insights into its relationship with atmospheric state variables. We also use the interpretations to identify the seasonality of the Madden-Julian Oscillation, and find that the conventionally defined extended seasons should be shifted later by one month. More generally, this study suggests that neural networks can be reliably interpreted for geoscientific applications

and may thereby serve as a dependable method for testing geoscientific hypotheses.

## 1 Introduction

Neural networks have the potential to improve our understanding of the earth system in ways that are unique from other statistical and machine learning methods. Recent research within the geosciences has shown that neural networks can be used to accelerate climate model parameterizations (Brenowitz and Bretherton, 2018; Rasp et al., 2018), discover patterns of earth-

system variability (Toms et al., 2019), and make accurate global weather predictions (Weyn et al., 2019), among numerous other applications in weather and climate (e.g., Barnes et al., 2019; Ebert-Uphoff and Hilburn, 2020). These advances have been rooted in the theory that neural networks are universal function mappers – that is, given a sufficient level of neural network complexity and quality of input data, a neural network can map any relationship between two datasets (Chen and Chen, 1995).





Neural networks may be particularly useful within the geosciences if the relationships contained within their learned param-
eters can be understood and interpreted. Numerous methods have been proposed for such interpretation within the computer
science community, and have even been shown to be applicable to improving the understanding of geoscientific phenomena
such as ENSO, sources of seasonal predictability, and severe convective storms (Toms et al., 2019; McGovern et al., 2019;
Gagne II et al., 2019; Ebert-Uphoff and Hilburn, 2020). The critical caveat of using interpretable neural networks within geo-
science is that the interpretations must accurately portray the relationships captured by the neural network and not mislead
the scientist toward incorrect conclusions. Therefore, any interpretability methods should first be tested on topics that are well
understood so that trust can be lent to studies that use the methods to discover entirely new patterns.

The Madden-Julian Oscillation (MJO; Madden and Julian, 1971; Wheeler and Hendon, 2004) has been a focus of hundreds
of publications across numerous decades, and although it is not fully understood, a few of its characteristics are commonly
accepted by the scientific community. For example, its core characteristic is an anomaly in deep convection and associated cloud
cover within the tropics that forms within the western tropical Indian Ocean and propagates eastward toward the tropical eastern
Pacific ocean over the course of 3 0 to 60 days (Hendon and Liebmann, 1994; Wheeler and Hendon, 2004; Kiladis et al., 2005).
While we will focus on the tropical characteristics of the MJO, it is also generally accepted that the atmospheric response to
deep convective heating within the MJO can generate teleconnection patterns across the globe (e.g., Roundy et al., 2010; Tseng
et al., 2019; Toms et al., 2020). The formation and propagation of the MJO are not as well understood, although numerous
theories have been put forth (Zhang et al., 2020), one of which suggests that the MJO propagates in response to gradients of
tropical water vapor anomalies (Sobel and Maloney, 2013; Adames and Kim, 2016). Another theory suggests that the MJO
could be a large-scale envelope of eastward and westward propagating gravity waves, and that its eastward propagation occurs
because the eastward waves travel faster than the westward waves (Yang and Ingersoll, 2011, 2013). Anomalies in atmospheric
state variables that coincide with the MJO are also well documented (Kiladis et al., 2005; Adames and Wallace, 2014; Monteiro
et al., 2014; Adames and Wallace, 2015), although their relationship with the seasonality of the MJO is less clear, particularly
given remaining uncertainties in mechanisms driving the seasonality of the MJO itself (Zhang and Dong, 2004; Jiang et al.,
2018).

We use the MJO as an opportunity to test whether interpretable neural networks can capture known patterns of variability
within complex geoscientific data, and we then extend our analysis into inferring new information about the MJO itself. We
also provide a new definition of MJO seasonality, for both the conventional outgoing longwave radiation definition and across
atmospheric state variables. The aim of this paper is threefold: 1) to highlight the ability of neural networks to capture complex
relationships within geoscientific data; 2) to test neural network interpretation methods to ensure they can reliably infer the
relationships captured by neural networks; and 3) use the interpretations to gain new insights into the MJO. This paper thereby
offers a conceptual guideline for how a geoscientist might go about using a neural network to discover new patterns within
geoscientific data. Those interested in the MJO itself will also find new insights into its spatial structures and seasonality.



## 2 Data and Methods

We first discuss the data we use to define the MJO and then detail how we design a neural network to infer information about its spatial structure and seasonality.

### 2.1 Data

We define the MJO according to the Outgoing Longwave Radiation MJO Index (OMI; Kiladis et al., 2014), which tracks the state of the MJO using anomalies in top-of-atmosphere outgoing longwave radiation (OLR; Liebmann and Smith, 1996). Increased cloud-cover inhibits the upwards ventilation of longwave radiation to space, so outgoing longwave radiation is generally used as a proxy for cloud cover in studies of the MJO. Some of the details of OMI are listed below, and it can generally be defined as a linear representation of the MJO based on outgoing longwave radiation anomalies with periods of

20 to 96 days. An important advancement of OMI beyond other MJO indices is that the structure of the MJO is calculated for each day of the year across a 121-day rolling window, and thereby accounts for seasonality. The index is constructed by calculating the two leading principal components in tropical (20°S to 20°N) outgoing longwave radiation anomalies, following the removal of the seasonal cycle and filtering the outgoing longwave radiation field to contain only eastward propagating waves with a periodicity of 30 to 96 days. The MJO also exhibits higher frequency modes of variability and occasional

westward propagation (Roundy and Frank, 2004; Zhao et al., 2013), so outgoing longwave radiation anomalies that include both eastward and westward propagating waves with periods of 20 to 96 days are then projected onto the 30- to 96-day principal components. This projection results in OMI including all eastward and westward propagating components of the MJO with periods of 20 to 96 days, with the caveat that they must coincide with the dominant, eastward propagating, 30- to 96-day mode of the MJO.

While the process of calculating OMI is complicated, the resultant phase-space and spatial perspectives of the MJO are relatively simple, as shown in Figure 1. A two-dimensional phase space is commonly used to define the phase and amplitude of the MJO, with each axis representing the two OMI principal components. As the MJO progresses, it completes a circle about its two-dimensional phase space, which represents the eastward propagation of a spatially coherent dipole in outgoing longwave radiation anomalies (Figure 1a). The phase space is conventionally separated into eight octants for convenience, so

the MJO is commonly studied according to its evolution across eight discrete phases. The phase of the MJO is based on the azimuth of the linear combination of the two principal components, and its magnitude is determined based on the distance of this point from the origin. An MJO event is generally considered to be "active" once the principal component magnitude is greater than 1, which is delineated by the red dots in Figure 1b. Because the principal components are standardized to have zero-mean and unit variance, MJO events of increasing amplitude become increasingly rare, such that most events have low

amplitude and are clustered about the origin.

We test whether a neural network can identify the phase of the MJO given inputs of cloud characteristics and atmospheric state variables. The inputs to the neural network are tropical (30°S to 30°N), 20- to 96-day filtered fields of outgoing long-wave radiation and 850-hPa, 500-hPa, and 200-hPa zonal wind, meridional wind, temperature, water-vapor mixing ratio, and

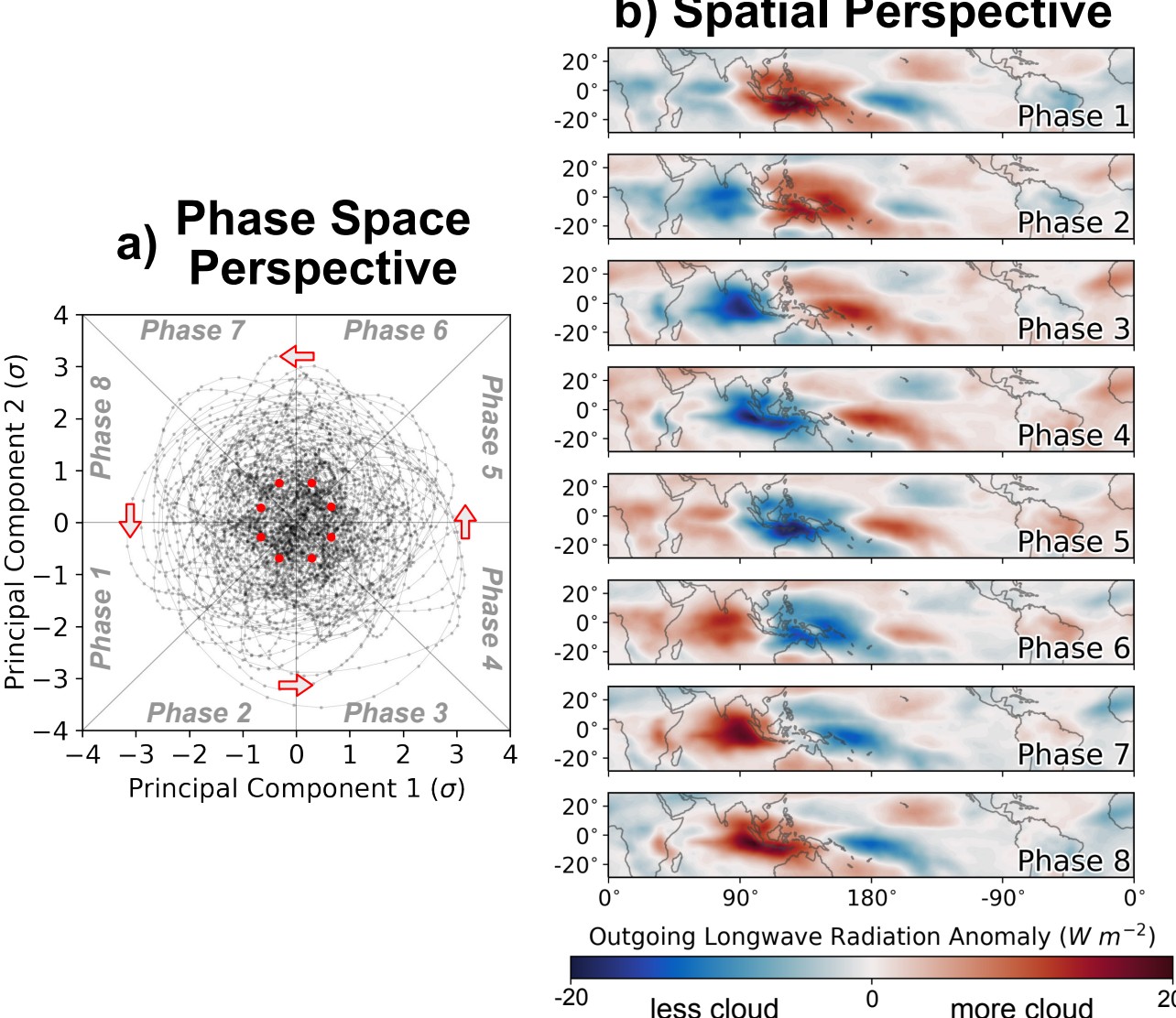

**Figure 1.** Spatial and phase-space perspectives of the Madden-Julian Oscillation. a) The phase space depiction of the MJO, again according to OMI for all MJO cases from January 1, 1980 through December 31, 2016.; (b) The spatial evolution of the MJO through its eight-phase phase space according to the outgoing longwave radiation MJO index (OMI)





geopotential (Figure 2a), and the outputs are the eight discrete phases of the MJO according to OMI. Only days during which

the MJO was active are used (i.e. its principal component magnitude was greater than one). We use atmospheric state variables from the NASA MERRA-2 reanalysis (Gelaro et al., 2017) and outgoing longwave radiation from the NOAA once-daily outgoing longwave radiation climate data record (Lee, 2014), both spanning from January 1, 1980 through December 31, 2016. We remove the seasonal cycle, defined as the annual-mean cycle from all 37 years of input data, before applying a 20- to 96-day Lanczos bandpass filter with 121 weights and interpolating each variable onto a homogeneous 2° grid. The training data

spans from January 1, 1980 through December 31, 2009, and the validation data span from January 1, 2010 through December 31, 2016. The training and validation data generally capture similar phase and amplitude distributions across each MJO phase (Figure 2b).

## 2.2 Neural Network Design

We design a neural network to be as simple as possible, while still ensuring it can capture any relationships between the input

atmospheric state variables and the phase of the MJO. We use fully-connected networks, which can be thought of as a chain of nonlinear regression functions that map the relationships between input and outputs datasets. The neural network has one input layer, two hidden layers with 64 and 128 nodes each, and one output layer with eight nodes, each of which represent a phase of the MJO (Figure 3). The hidden nodes all use the ReLu activation function, which applies the $max(0, x)$ operator to the output of each node. A softmax operator is applied to the output layer, which normalizes the output of the neural network

such that the sum across all output nodes is equal to one. The outputs can therefore be thought of as a likelihood, with higher values for each node corresponding to a higher likelihood that the input sample belongs in that particular phase of the MJO. During labeling, each MJO event is labeled using an eight unit vector, and each unit represents one phase of the MJO. An input associated with phase three of the MJO would therefore have an output label of [0,0,1,0,0,0,0,0], which in the perspective of the neural network implies a 100% likelihood that the sample is associated with phase 3 of the MJO.

We train separate neural networks on data from 121-day bins centered on each calendar week of the year in order to study the seasonality of the MJO. Each neural network is therefore tasked with identifying the phase of the MJO according to the outgoing longwave radiation and state variable patterns during the period of the year to which it is assigned. Comparisons between interpretations of each neural network offer insights into the seasonality of the MJO, as discussed in subsequent sections.

Neural network interpretability generally becomes more challenging with increasing network complexity (Montavon et al., 2018). The neural network design we use is simple enough to enable robust interpretations, but complex enough to capture useful relationships between the input state variables and MJO phase. We find that decreasing the number of internal nodes reduces the accuracy, presumably because the network is then not complex enough to model the relationships between the atmospheric state variables and MJO. On the other hand, increasing the number of nodes also reduces the accuracy of the

neural network on the validation dataset, because it is able to overfit on meaningless noise within the inputs using the additional weights and biases. We address any overfitting by applying L2-regularization to the weights connecting the input layer to the first layer of hidden nodes, which forces the network to focus its attention on broader spatial patterns within the inputs. We



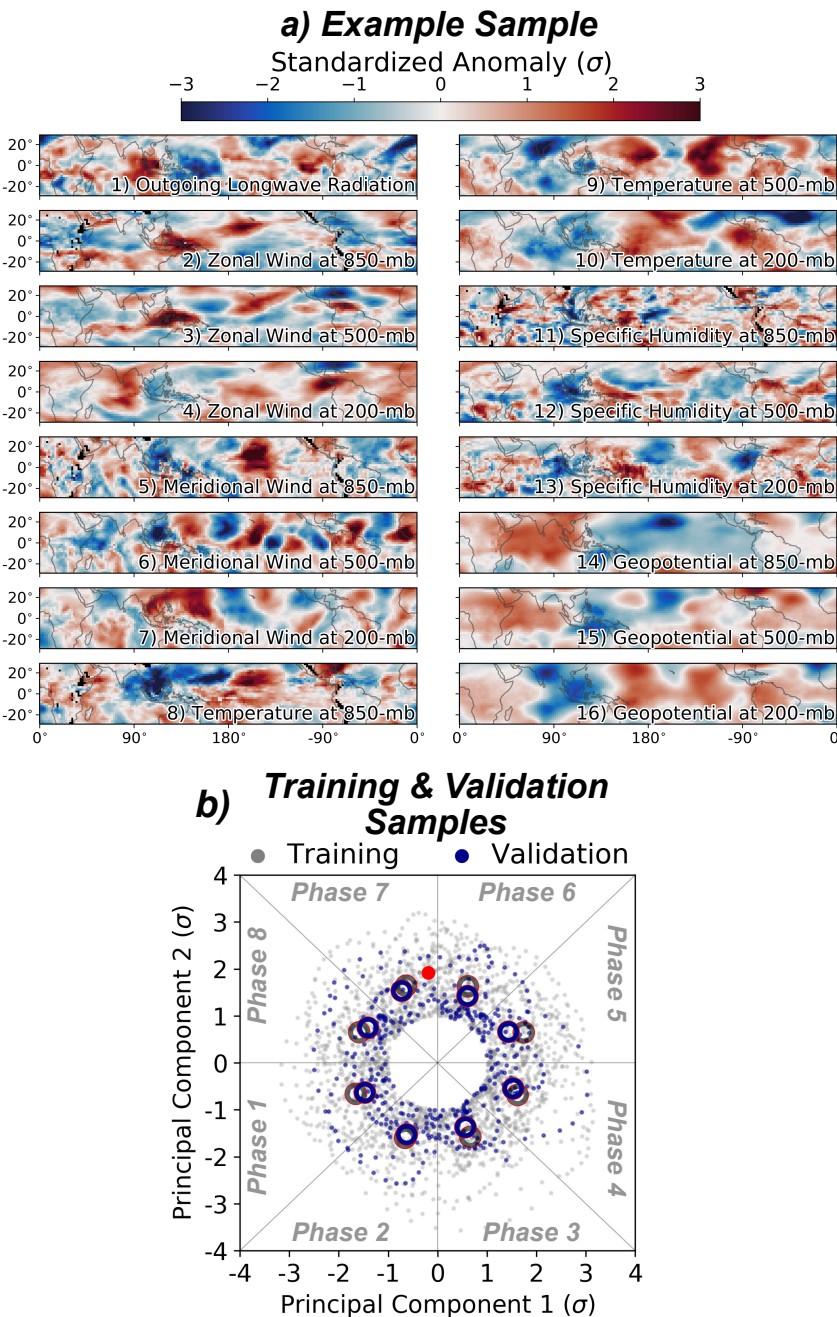

**Figure 2.** (a) An example input sample, which corresponds to a Phase 7 MJO day. Each variable was standardized for each grid point to have zero mean and unit variance across all samples from January 1, 1980 through December 31, 2016. (b) A visualization of how the samples are split between the training and validation datasets. The red dot corresponds to the sample shown in (a), the gray denotes denote the training samples, and the purple dots denote the validation samples. The gray rings denote the training sample mean phase and amplitude for each phase, and the blue rings denote the same but for the validation data.





# Neural Network Design for MJO Phase Identification

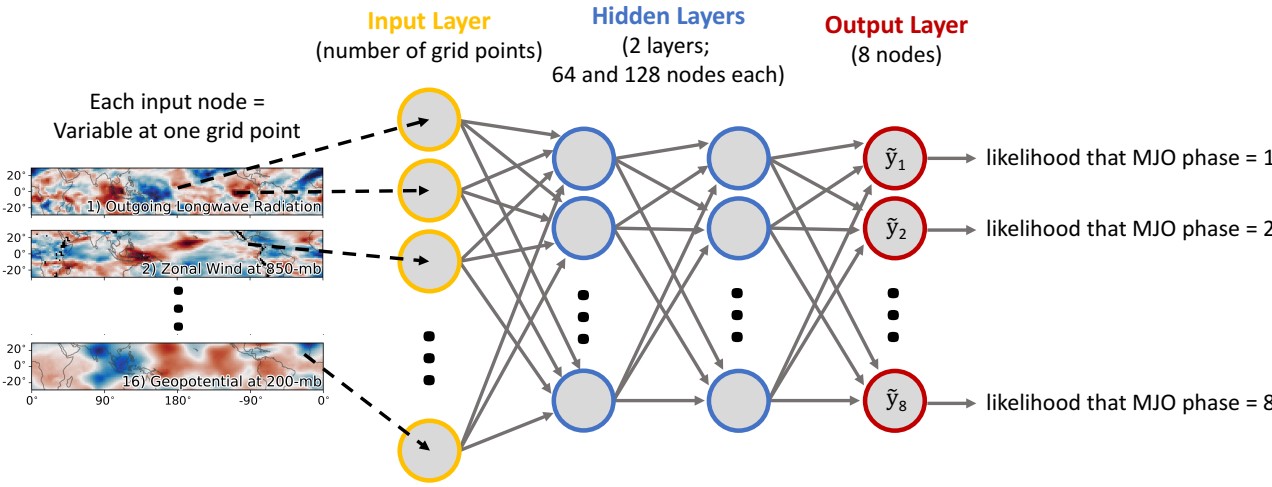

**Figure 3.** Schematic for the neural network used in this study. The first layer ingests vectorized input images, with two subsequent hidden layers the first with 64 nodes and the second with 128 nodes, and an output layer of 8 nodes that correspond to the eight phases of the MJO. A separate neural network is trained for each calendar week of the year.

thoroughly tested the accuracy of convolutional neural networks (CNNs) for our particular problem, and found that the fully connected networks were both more accurate and interpretable than their CNN counterparts when L2 regularization is applied

to the first layer of hidden nodes.

## 2.3  Neural Network Interpretability

The novelty of this paper is the demonstrated ability to interpret what the neural networks have learned, and to then gather scientific value from the interpretations. We use two interpretation methods which we briefly discuss here and are explained in more extensive detail in the context of geoscience within Toms et al. (2019). The two methods we use are called backward

optimization and layerwise relevance propagation, both of which map the decision-making process of the neural network onto the original input dimensions.

Backward optimization uses the same method that is used to train a neural network (i.e. backpropagation) to instead interpret what a trained network has learned (Simonyan et al., 2013; Yosinski et al., 2015; Olah et al., 2017). Rather than updating the weights and biases of the network, the input itself is updated to minimize the difference between the network's associated

output and a user-defined output. This process generates a single optimized pattern associated with a particular output, and thereby offers a composite interpretation of patterns contained within a neural network. In our case, we input blank (i.e. all-





zero) maps, and optimize them to be most closely associated with a particular phase of the MJO. In doing so, we can identify the optimal patterns in each state variable for each phase of the MJO.

Layerwise relevance propagation (LRP) interprets the neural network's decision-making process for each individual input

sample (Bach et al., 2015; Montavon et al., 2017, 2018). Given a trained neural network, an input sample is passed forward, the associated output is collected, and the unique pathways through which information flows from the input to the output for that specific sample are analyzed. The pathways are traced by propagating information backwards from the output layer to the input layer using rules specific to LRP. By tracing these pathways, the "relevance" of each input variable to the network's associated output can be quantified for each individual input example. The resultant relevance is unique to each input sample, because

the pathways through which information flows through a neural network is similarly unique for each sample. A particularly important aspect of LRP is that the formulation of neural network that we use (i.e. fully connected networks with ReLu activation functions) conserves the relevance from the output layer to the input layer, meaning that all information important to the network's decision is included within the final LRP interpretation. LRP traces the information that *positively* contributes to the output of the neural network, so is well suited to categorical output. So, in our case, LRP shows which regions of

atmospheric state variables are most relevant to increases in the neural network's confidence that the sample belongs to a particular phase of the MJO.

## 3 Results

### 3.1 Neural Network Accuracy

We first ensure the neural networks are accurate enough to offer scientifically valuable interpretations. As a reminder, we train

separate neural networks on data from 121-day windows centered on each calendar week of the year. The accuracy of the neural network for the window centered on January 10th is presented from both a deterministic and probabilistic perspective in Figure 4. The deterministic accuracy is assessed by counting the number of input samples the neural network assigns to the correct phase of the MJO. The most common error of the neural network is to assign an input sample to a phase that is one phase prior to or after the correct phase, which is likely caused by the MJO being a continuous phenomenon that we have discretized

for the sake of interpretation. So, another useful accuracy metric is how often the neural network correctly assigns the input samples into either the correct phase or one phase before or after the correct phase. For the neural network centered on January 10th, the deterministic accuracy without a one-phase buffer is 74% and the accuracy with a one-phase buffer is 92%. From the composite probabilistic perspective (Figure 4b), the neural network assigns the highest likelihoods to the correct phase, although the phases immediately before and after the correct phase also have appreciably high likelihoods.

An important question regarding the usage of neural networks is whether they out-perform conventional methods, such as regression. If regression performs similarly to a neural network, then the increased complexity and nonlinearity of a neural network is not required. We therefore similarly use linear regression to identify the phase of the MJO using the input state variables and outgoing longwave radiation across 121-day windows centered on each calendar week. The linear regression models have no hidden nodes and no nonlinearities, but are otherwise identical to the neural networks in that the regression

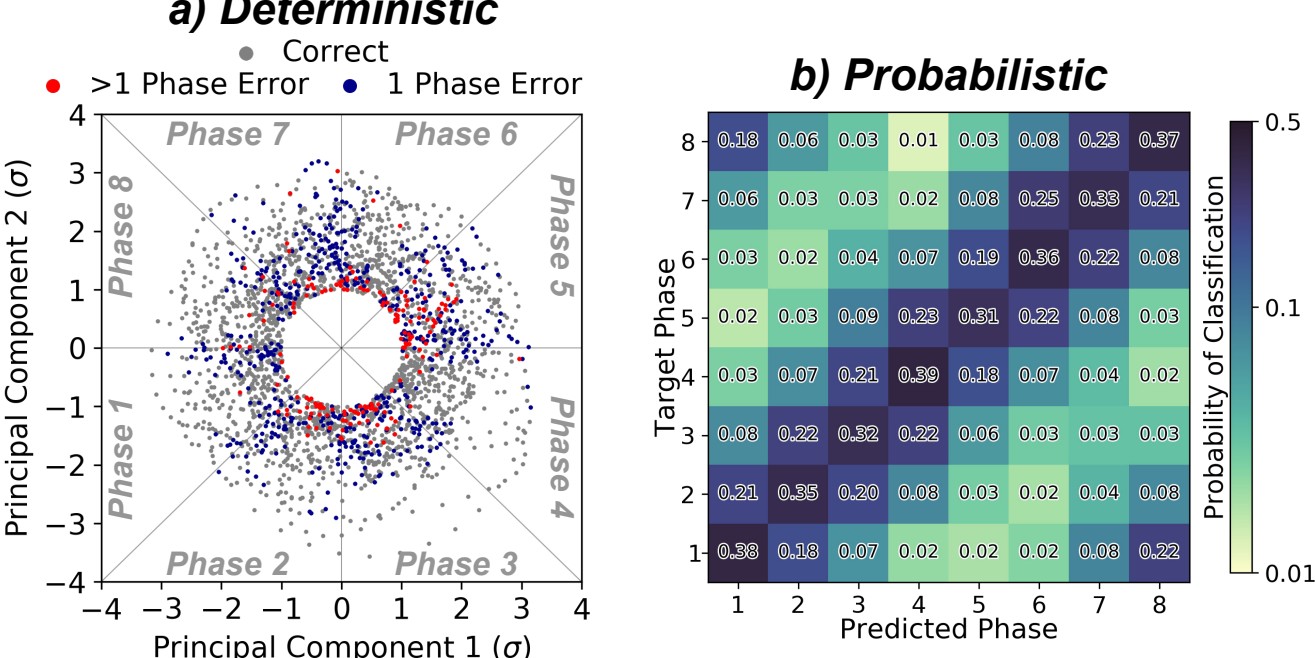

**Figure 4.** Example visualizations of the accuracy of the neural networks, in this case for the neural network centered on January 10. (a) Deterministic accuracy, where samples that are correctly classified are colored grey, those assigned to a phase one before or after the true phaes are colored blue, and those assigned to a phase two or more different from the true phase are colored red. (b) Probabilistic accuracy, where the average probabilities assigned to each sample within the validation dataset is shown for each target phase. The probabilities summed across each row sum to one.

model assigns a normalized likelihood that the input is associated with a particular MJO phase by using a softmax operator before the final output. We regularize the regression models using L2-regularization to ensure they are not overfit to the training data, similar to the neural networks. The accuracies of the neural network and linear regression approaches are compared in Figure 5. The neural networks are nearly twice as accurate as linear regression for all weeks of the year, which means the nonlinearities and increased number of pathways for information to flow through the neural network are essential to modeling 175 the spatial structures of the MJO. We therefore conclude that interpretations of the neural networks can offer insights into relationships between the MJO and atmospheric state variables that conventional linear methods can not.

**3.2 Interpreting the Neural Network**

**3.2.1 Identifying the Spatial Structures of the MJO**

We use backward optimization and layerwise relevance propagation (LRP) to infer the spatial structure of the MJO and its 180 seasonality according to the neural networks. Examples of LRP applied to inputs for the neural network trained on the 121-

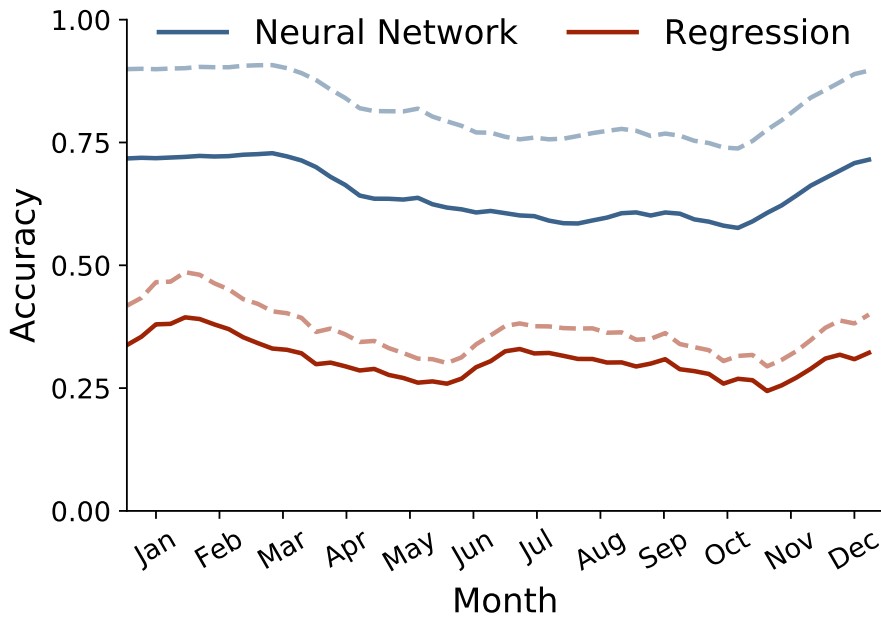

**Figure 5.** The accuracy of the neural network and linear regression approaches for each calendar week throughout the year. The neural network accuracy is plotted in blue, and the regression accuracy is plotted in red. The solid lines show the accuracy for all input samples, and the dashed lines show the accuracy if a one-phase error is permitted.

day window centered on January 10th are shown in Figure 6. We use four examples of MJO phase 7 for which the neural network correctly identifies the phase of the MJO, and for simplicity we only show the LRP maps for outgoing longwave radiation although similar maps are generated for each input variable. The LRP maps show that the neural network focuses its attention on outgoing longwave radiation anomalies across the Maritime Continent, particularly within its eastern extent, which

is consistent with previous research on the regions of convection associated with phase 7 of the MJO (Wheeler and Hendon, 2004; Kiladis et al., 2014). The LRP heatmaps also highlight the spatial uniqueness of each phase 7 MJO event, which can not be inferred by a linear regression model. It is likely that the increased accuracy of the neural networks compared to the linear regression models is caused by this ability of the neural network to capture the spatial uniqueness of each event.

We next test the neural networks more rigorously, and challenge them to identify the most common spatial structures of

the MJO across its eight phases. To do so, we use backward optimization, and optimize inputs such that the spatial patterns within the inputs make the neural networks most confident that the inputs are associated with a particular phase of the MJO. Numerically, this means that the outputs associated with the optimized inputs have a likelihood of approximately 1 in the phase for which they are optimized, and likelihoods of 0 for all other phases. We again only show the optimized outgoing longwave radiation fields for simplicity, although the optimization also identifies the characteristic patterns in the 15 other state variables.

The spatial pattern of the MJO during boreal winter (January 10th) and boreal summer (August 1st) according to both OMI and the neural networks are shown in Figure 7. The neural networks capture similar features to OMI during both seasons,

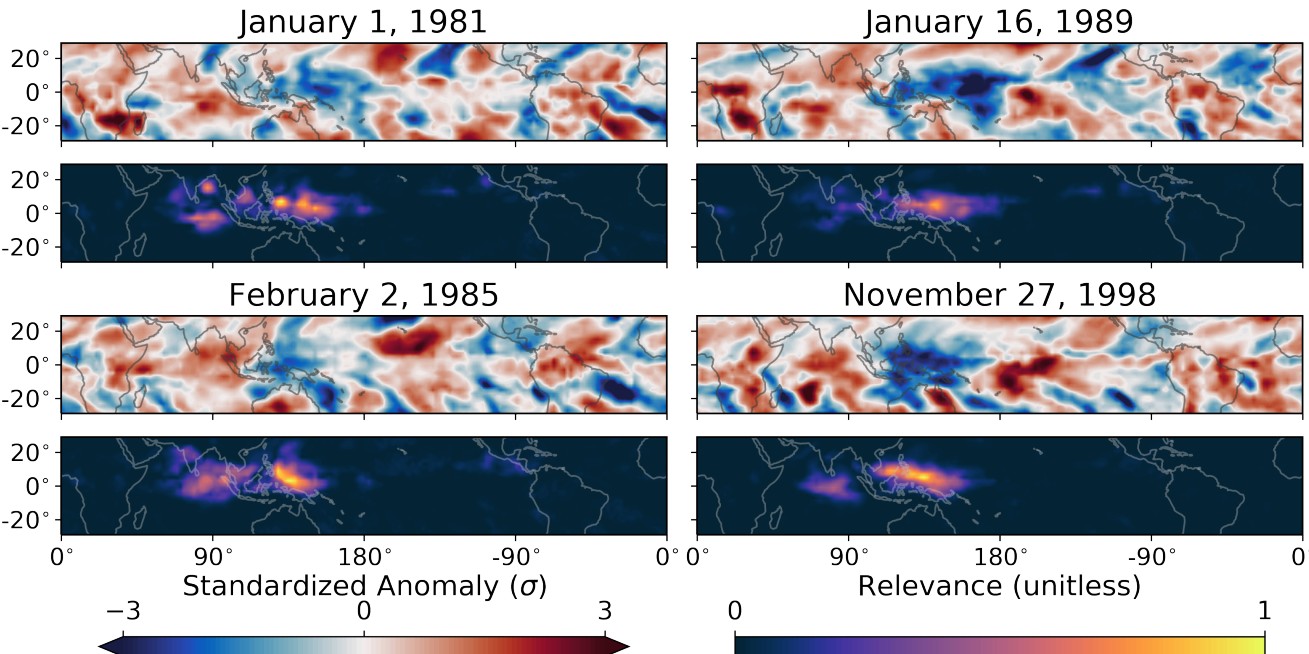

**Figure 6.** Example relevance heatmaps from the layerwise relevance propagation interpretation technique. The outgoing longwave radiation field from four example inputs into the neural network are shown, each corresponding to a separate Phase 7 MJO day. The corresponding relevance heatmaps are shown below each example outgoing longwave radiation field, and shows where the neural network focuses its attention to determine that the examples are associated with a Phase 7 MJO day.

in particular the prominent eastward propagation during boreal winter and the transition to northeastward propagation during boreal summer. The neural network focuses on a core of outgoing longwave radiation anomalies across the Indo-Pacific region and within the eastern Pacific, while OMI includes a greater magnitude of anomalies within the central Pacific. Given the similarities between OMI and the composite neural network interpretations, we conclude that nonlinearities of the MJO are primarily manifested through the uniqueness of each event as highlighted in Figure 6. The composites of LRP relevance similarly capture the dominant structures of the MJO during both seasons, and agree rather well with the optimized inputs (Figure 7).

### 3.2.2 Testing the Seasonality of the MJO

Because the neural network so accurately captures the seasonal evolution of the MJO within the outgoing longwave radiation composites, we now extend the interpretations to study the seasonality of the MJO. We first test how the spatial structure of the MJO changes across seasons using LRP. To do so, we calculate the composite relevance for each variable for each calendar week of the year, and present the annual evolution of the relevance in Figure 8. The relevances of each variable exhibit unique

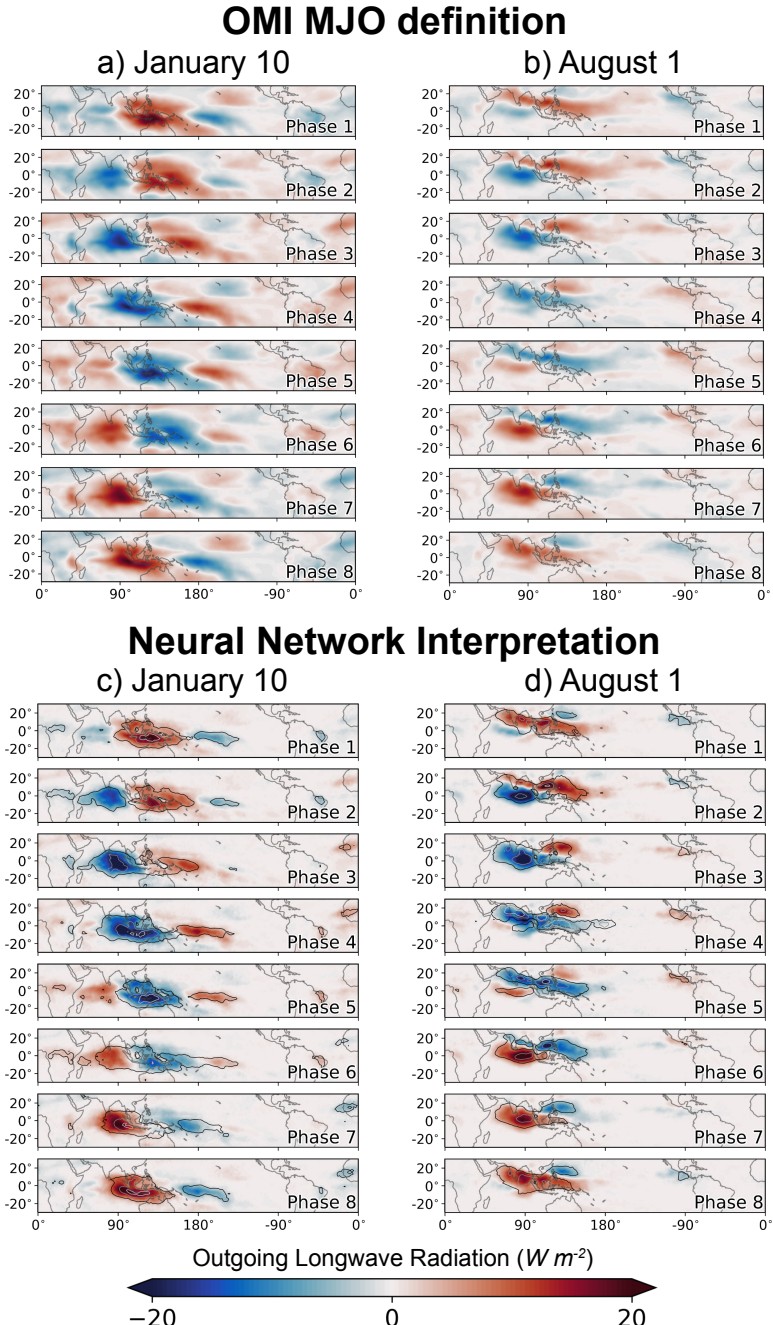

**Figure 7.** (a, b) The outgoing longwave radiation fields for each MJO phase according to the OMI for the boreal winter (January 10) and boreal summer (August 1) examples and those identified by the neural network. (c, d) The outgoing longwave radiation fields for each MJO phase according to the neural network based on the backward optimization and layerwise relevance propagation interpretation methods. The fill value shows the optimized outgoing longwave radiation patterns for each phase of the MJO, and the open contours show the composited relevance from LRP for all samples within each phase.

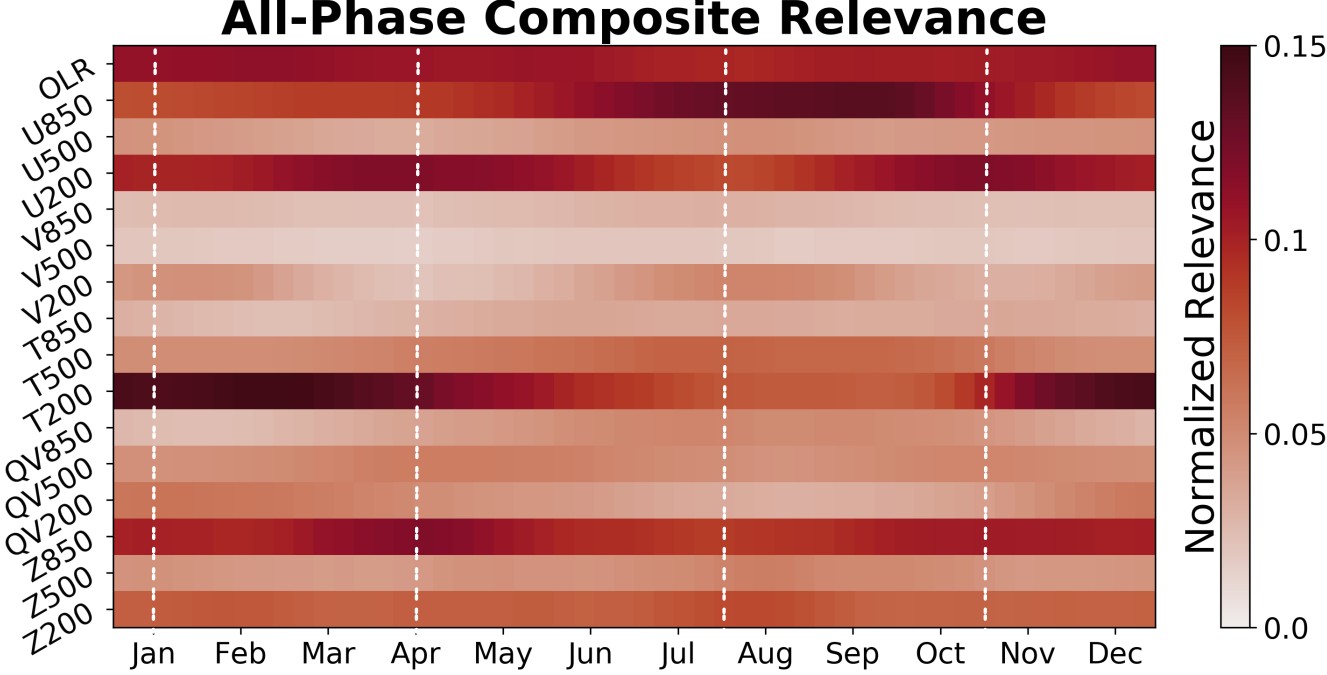

**Figure 8.** Composite normalized LRP relevance across all variables for each calendar week throughout the year. The relevance is normalized to sum to one across all variables for each calendar week (i.e. along the vertical axis).

seasonal cycles, aside from outgoing longwave radiation which is similarly relevant throughout all periods of the year. For
example, the seasonal cycle of lower-tropospheric zonal wind (U850) reaches a maximum in relevance during boreal summer, whereas upper-tropospheric zonal wind (U200) is most relevant during the spring and fall. Some variables exhibit a uni-modal seasonal cycle (e.g. U850, T200), whereas other variables exhibit a bimodal seasonal cycle (e.g. U200, V200, Z850) In general, upper-tropospheric anomalies are most relevant during boreal winter, while lower-tropospheric anomalies are most relevant during boreal summer.

The fact that upper-tropospheric anomalies are most relevant to the MJO during boreal winter may explain the seasonality in coupling between the MJO and the stratosphere (Son et al., 2017; Densmore et al., 2019; Toms et al., 2020). Previous research has hypothesized that the MJO can be modulated by sources of stratospheric variability such as the quasi-biennial oscillation through a downward influence of upper-tropospheric temperature anomalies (Abhik and Hendon, 2019; Martin et al., 2020). So, because upper-tropospheric thermodynamic anomalies are particularly relevant to the MJO during boreal winter (Figure 8),
then any influences on the thermodynamic structure of the upper troposphere by the stratosphere may have an increased impact on the MJO. This discussion highlights the capability of neural network interpretations to guide and test proposed hypotheses, although a direct test of this hypothesis is beyond the scope of this paper.







**Figure 9.** Optimized patterns for phase 6 of the MJO for different periods of the year. The central date on which the neural network is trained for each optimization is shown in the title of each subfigure. Each subfigure shows outgoing longwave radiation, 850-mb zonal wind, 200-mb zonal wind, 200-mb meridional wind, 200-mb temperature, and 850-mb specific humidity.





We now examine the optimal spatial patterns of the MJO throughout the year to provide some spatial context to the seasonality of the relevances shown in Figure 8. Figure 9 shows the optimal spatial patterns associated with phase 6 of the MJO at four
different times of the year, the times of which are denoted by the dashed white lines in Figure 8. In general, the spring structure of the MJO is more similar to the winter structure than the summer structure. The April 15 optimal patterns are nearly identical to the January 15 optimal patterns, aside from lower-tropospheric moisture anomalies which are more similar between April 15 and August 1. The upper-tropospheric anomalies during boreal winter are more representative of a Matsuno-Gill type response to convective heating (Matsuno, 1966), whereas during boreal summer the signature is more diffuse and elongated across the
equator. Figure 9 is generally supportive of the idea that lower-tropospheric anomalies are most relevant during boreal summer whereas upper-tropospheric anomalies are most relevant during boreal winter.

Mechanistic studies of the MJO commonly depend on accurate definitions of when each MJO seasonal mode occurs, since the spatial structures of the winter and summer modes differ so substantially (Figure 9). Should the seasonal definitions of the MJO be inaccurate, then there is a risk that the mechanistic studies themselves are not targeting processes specific to each
season. We therefore use the backward optimization interpretations of the neural networks to define the MJO seasonal modes. To do so, we spatially correlate the optimal patterns for each state variable to the optimal patterns on January 10th and August 1st, which are generally considered to be the peak of the boreal winter and boreal summer modes. We then define the boreal winter mode to exist during periods for which the optimized MJO patterns have a correlation of greater than 0.75 with the optimized pattern for January 10th, and similarly define the boreal summer mode to exist when the correlation is greater than
0.75 with the optimized pattern for August 1st. Using our definition, the seasonality differs across atmospheric state variables, although the boreal winter and summer modes generally span from late November through early March and early June through early October, respectively (dark colors in Figure 10). Lower-tropospheric variables generally lead the transition from the boreal winter mode to the equinoctial transition toward the boreal summer mode, although a less clear relationship exists during the transition back to the boreal winter mode.

Finally, we define extended boreal winter as the period during which the correlation between each weekly optimal pattern and the January 10th optimal pattern is greater than that between the weekly optimal patterns and the August 1st optimal pattern. Extended boreal summer spans the rest of the year. Using this definition, extended boreal winter MJO extends from early November through late April across most state variables, and from mid-November through late April for outgoing longwave radiation in particular (dark and light colors in Figure 10). Many studies of the MJO have previously used extended winter and
summer seasons which span the months of October through March and April through September, respectively (Yoo and Son, 2016; Zhang and Zhang, 2018). Our results suggest that these extended seasons should, at a minimum, be shifted one month later in the year.

## 4   Discussion and Conclusions

We have tested the ability of interpretable neural networks to identify complex, multi-scale geophysical phenomena via their
application to the Madden-Julian Oscillation (MJO). We first evaluated whether neural networks can identify the MJO, and

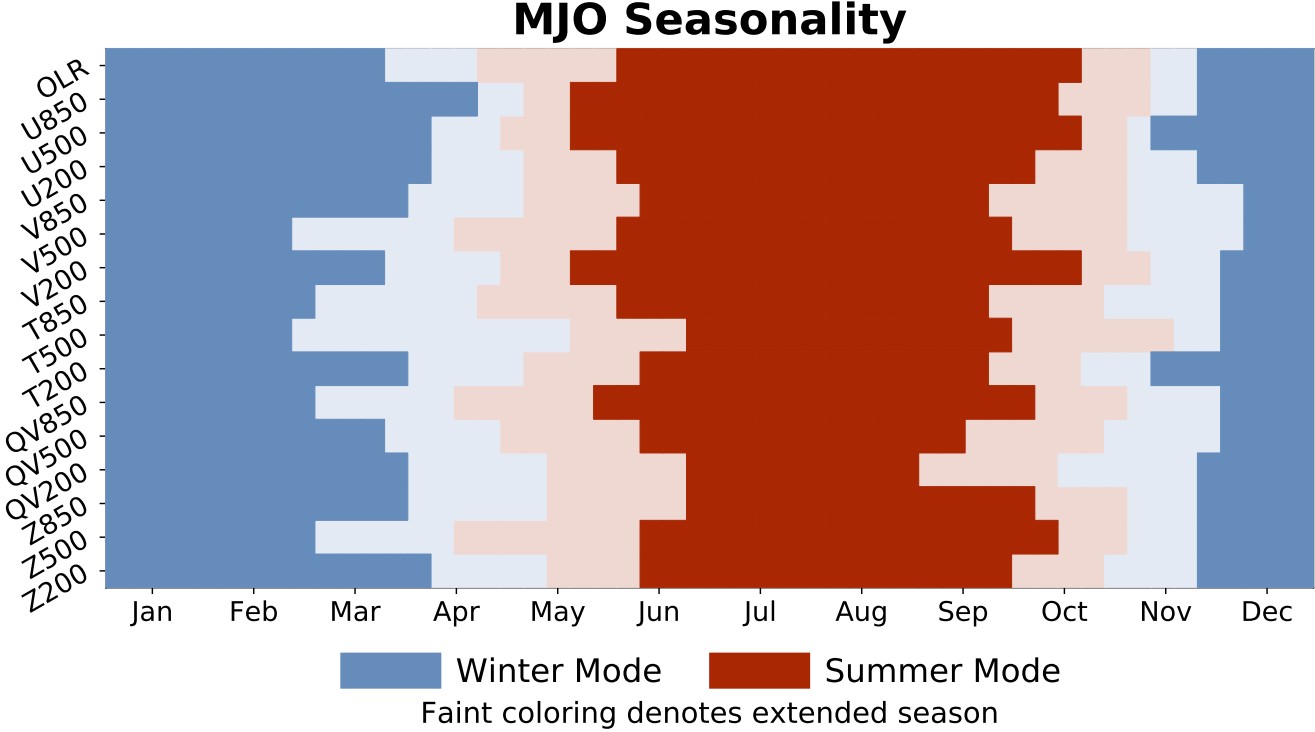

**Figure 10.** Seasonality of the Madden-Julian Oscillation according to interpretations of the neural networks. The extended boreal summer and winter modes are shown in red and blue, respectively, and periods of transition are denoted by the lighter red and blue colors. The winter (summer) mode is defined as periods during which the correlation between the optimized MJO pattern on January 10th (August 1st) and the optimized pattern for each respective calendar week is greater than 0.75. and the transition periods extend between these two modes. The extended boreal winter mode is defined as periods during which the optimized pattern for each respective calendar week is more highly correlated with the January 10th optimized pattern than the August 1st optimized pattern, and visa versa for the extended boreal summer mode.

then used neural network interpretability methods to study the seasonality and spatial structure of the MJO and its relationship to atmospheric state variables. Our study therefore contributes both to the general usage of neural networks within geoscience and to knowledge of the MJO itself, so we separate our discussion of the implications for both communities below.

## 4.1 Implications for Neural Networks in Earth Science

We have shown that neural networks are highly interpretable, even for complex, multi-scale geophysical phenomena. Two methods proposed by the computer science community – backward optimization and layerwise relevance propagation – provide particularly useful interpretations of neural networks (Toms et al., 2019). Namely, backward optimization offers composite interpretations, while layerwise relevance propagation enables interpretations on either a composite or case-by-case basis.





Both methods project the decision-making process of a neural network back onto the original dimensions of the input, which is
particularly useful for geoscientific applications where each input variable may have unique physical importance to the problem
being studied.

The capability of neural networks to include nonlinearities and simultaneously model different input patterns that lead to
similar outputs proved useful for studying the seasonality of the MJO. The neural networks identified the phase of the MJO
twice as accurately as linear regression, which implies that interpretations of the neural network characterize the MJO more
accurately than linear regression. We hypothesized that the increase in accuracy was caused by the neural networks' ability to
model the uniqueness of each MJO event, which is not feasible using conventional linear approaches such as regression. The
amount of neural network complexity required for tasks across the geosciences will vary greatly, so the benefits of interpretable
neural networks are also likely to vary across sub-disciplines. We have found that a baseline approach of comparing the
accuracy of neural networks to more simple methods such as linear regression is useful in determining the necessity of a neural
network.

Based on this study and other supporting work (Toms et al., 2019), the interpretations of what a neural network learns can
be used to advance geoscientific knowledge. Even for cases where interpretability is not the main objective, neural network
interpretations can offer insights into how and why neural networks are making their decisions, and can be used to ensure
that neural networks are making decisions using reasoning consistent with physics. While we use a relatively simple type of
neural network, the proposed methods are applicable to other types of neural networks as well, such as convolutional neural
networks and long short term memory (LSTM) networks. We found fully connected networks to be particularly useful for our
application and more accurate than convolutional neural networks, which, in light of the surging popularity of convolutional
neural networks within geoscience, suggests that fully-connected networks also have utility for geospatial problems.

### 4.2 Implications for the Madden-Julian Oscillation

We also used neural networks as an approach to better understand the spatial structure and seasonality of the MJO. Our results
are generally consistent with the thorough body of literature on the MJO, which supports the reliability and robustness of
interpretable neural networks within geoscience.

Consistent with previous studies, we find that the spatial structure of the MJO generally exhibits two dominant modes
of variability distinguished between the boreal summer and winter. We find that the extended boreal winter mode of the
MJO occurs between early November and late April, with the boreal summer mode occurring throughout the remainder of
the year. This definition of the extended seasons is delayed one month compared to conventional definitions, which use an
extended boreal winter of October through March. Furthermore, the seasonality of the relationship between the MJO and
atmospheric state variables is more complex, with each variable exhibiting a unique seasonality. Some state variables such as
lower-tropospheric zonal winds exhibit a uni-modal seasonality, whereas others such as upper-tropospheric zonal winds exhibit
a bi-modal seasonality. We also find that upper-tropospheric thermodynamic anomalies are particularly relevant to the MJO
during boreal winter, which may relate to the enhanced coupling between the MJO and stratospheric processes during this
season.





Consistent with previous studies, we find that the spatial structure of the MJO generally exhibits two dominant modes of variability distinguished between the boreal summer and winter. We also extend our analysis to test numerous aspects of the MJO, from its nonlinearities to its relationships with atmospheric state variables. The key points of this analysis are as follows:

1. The neural networks identify the phase of the MJO twice as accurately as linear regression, which suggests that non-linearities are important to the structure of the MJO. These nonlinearities are reflected in the spatial uniqueness of each MJO event, given that the composite structure of the MJO identified by the neural networks and linear methods are remarkably similar (Figure 5; Figure 6).

2. Each state variable exhibits a unique seasonality in its relationship with the MJO. For example, some state variables such as lower-tropospheric zonal winds exhibit a uni-modal seasonality, whereas others such as upper-tropospheric zonal winds exhibit a bi-modal seasonality (Figure 8; Figure 9).

3. Upper-tropospheric thermodynamic anomalies are particularly relevant to the MJO during boreal winter, which may relate to the enhanced coupling between the MJO and stratospheric processes during this season (Figure 8).

4. We find that the extended boreal winter mode occurs between early November and late April, while the boreal summer mode occurs throughout the remainder of the year. This definition of the extended seasons is delayed one month compared to the conventional definition, which uses an extended boreal winter of October through March (Figure 10).

Our results show that neural networks are highly interpretable, even for spatially complex geoscientific applications. Because of the high reliability of the interpretations, neural networks are viable tools for testing hypotheses related to the MJO and other spatially complex geophysical phenomena. More complex hypotheses can now be tested: for example, does horizontal advection of the lower-tropospheric mean moisture by the MJO circulation govern the propagation of the MJO (e.g., Jiang et al., 2018)? Or, a neural network can be used to identify whether an MJO event will initiate given spatial inputs of atmospheric variables, from which interpretability methods can identify the most relevant patterns for MJO initiation. A critical requirement for using neural networks in such studies is the proven ability to reliably interpret what the networks have learned, which is now possible.

*Author contributions.* BT, KK, P, and DY conceived the idea and designed the experiment. KK and P provided expertise on neural networks throughout the project. BT performed the analysis and wrote the paper.

*Competing interests.* The authors declare no competing interests.

*Code and data availability.* All data used in this study and an example script for training a neural network and generating the LRP heatmaps and optimal input fields is available at the following DOI: https://doi.org/10.5281/zenodo.3968896. The unprocessed versions of the MERRA-



2 reanalysis data used in this study are available for download at the National Aeronautics and Space Administration (NASA) Goddard Space Flight Center (GSFC) Global Modeling and Assimilation Office website: https://gmao.gsfc.nasa.gov/reanalysis/MERRA-2/. The unprocessed versions of the NOAA once-daily outgoing longwave radiation climate data record is available for download from the following website: https://www.ncdc.noaa.gov/cdr/atmospheric/outgoing-longwave-radiation-daily. The unprocessed MJO OMI data is available from the Na-

330   tional Oceanic and Atmospheric Administration Physical Sciences Laboratory (NOAA PSL) at the following website: https://www.psl.noaa .gov/mjo/mjoindex/.

*Acknowledgements.* This work was supported by the Department of Energy Computational Science Graduate Fellowship via grant DE-FG02-97ER25308 (B. T.). This work was also supported by the U.S. Department of Energy, Office of Science, Office of Biological and Environmental Research, Climate and Environmental Sciences Division, Regional  Global Climate Modeling Program, under Award DE-

335   AC02-05CH11231, the Laboratory Directed Research and Development (LDRD) funding from Berkeley Lab, provided by the Director, Office of Science, of the U.S. Department of Energy under Contract DE-AC02-05CH11231, the National Institute of Food and Agriculture, under the project CA-D-LAW-2462-RR, and the Packard Fellowship for Science and Engineering (D. Y.).



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
