# Peer review of "Testing the Reliability of Interpretable Neural Networks in Geoscience Using the Madden-Julian Oscillation"

_Geoscientific Model Development, 2020_

## Referee Comment (RC1) · Anonymous Referee #1 · 10 Aug 2020

The authors apply a neural network to project the phase of the real time multivariate MJO index from gridded data. They present the network with fields of data and the corresponding phase numbers in the index to train the network to predict the phase from the data. They compare results against those achieved by comparison against a similar linear regression model. The result is well presented, and, were it not for an important weakness, I would recommend publication. The normal pathway to detecting the phase from the data is to simply project the data onto the EOF patterns associated with the leading two eigenmodes in the combined data matrix of zonal wind and OLR anomalies, then to label the phases as segments of the phase space generated by PC1 and PC2. The network evidently can replicate the results of the projection

approach well, but the comparison against the linear regression approach is unfair. Although the authors used a full grid of data for their demonstration, I used PC1 and PC2 (together with a column of ones) to predict the phase numbers using linear regression as a demonstration of concept. The fundamental weakness of linear regression emerges immediately. It is that phase number goes from 1 to 8, and phases 1 and 8 are proximate to each other (that is, phase is a cyclic variable). That is, in terms of the comparison between neighboring phases, phases 8 and 1 are just as proximate to each other as phases 4 and 5, in the phase space, but they are as far apart as any two phases can be, in terms of the phase number. Thus following the linear regression approach, phases 4 and 5 will be predicted well by linear regression, but the lowest & highest phases will have large errors because the regression approach cannot yield a linear model that disaggregates signal that projects well onto phases 8 and 1 both. Thus much of the advantage of the neutral network may be in being able to address a signal that is *defined* as nonlinear from the start. If, in contrast, the authors had applied both the NN and the regression model to predict the OLR anomaly at a given grid point or region (a relationship that could have a large linear component), it is not so clear that the neural network would do better. The authors result in comparison against linear regression seems to arise simply because they defined a nonlinear frame for comparison, which is, phase number. It is trivially obvious that the linear regression cannot replicate that point. I think the authors could quickly verify that linear regression does well during phases 4-5, but poorly in 7,8, 1,2. These errors near phases 1 and 8 will dominate the difference between the linear regression and neural network. The problem, then, is that the authors chose a context in which linear regression cannot work. Yet there are many other contexts in which there might be a fair comparison. For example, using the NN and linear regression to predict OLR anomalies in a given region based on the PCs.

---

## Author Comment (AC1) · 18 Aug 2020

Authors' response to anonymous referee 1; GMD 2020-152

We appreciate the quick review from the first anonymous referee. It seems the referee misunderstood our methodology, and so we seek to clarify this misunderstanding below.

The reviewer's comment originates from our usage of the term "linear regression", which the reviewer seems to have interpreted differently than how we implement the method in our paper. In the paragraph starting on Line 165, we begin a discussion

of how we approach a comparison between a nonlinear neural network and a linear regression-like approach. In this paragraph, we state "the linear regression models have no hidden nodes and no nonlinearities, but are otherwise identical to the neural networks in that the regression model assigns a normalized likelihood that the input is associated with a particular MJO phase by using a softmax operator before the final output". While this is a fairly wordy sentence, the key point here is that we use eight linear regression models, each connected from the input values to an output node associated with a particular phase of the MJO. We then identify the maximum value across these eight output nodes and accept the corresponding phase as the predicted phase. This type of model is effectively similar to a neural network, except that the neural network has no hidden layers and so is not permitted to use any nonlinearities. The method we use is a more complex version of linear regression than the rather simple approach that the reviewer suggests we used. Our approach is described schematically in the attached Response Figure 1 (the associated caption is pasted at the end of this response).

Given this misunderstanding, the reviewer's proposed method of using only PC1 and PC2 to identify the phase of the MJO using a conventional linear regression technique is an over-simplification of the method that we use. We therefore agree with the reviewer that their proposed method will not work well, and we therefore used a more complex form of linear regression in our original manuscript. To address this misunderstanding, we propose the following changes to the wording within our manuscript, which we will implement once the discussion period has closed:

1) Throughout the manuscript, our usage of the phrase "linear regression" will be changed to "multi-output linear regression"

2) The rather complicated sentence explaining how we implement multi-output linear regression will be changed from (as written in Lines 168 through 171) "the linear regression models have no hidden nodes and no nonlinearities, but are otherwise identical to the neural networks in that the regression model assigns a normalized likelihood that

the input is associated with a particular MJO phase by using a softmax operator before the final output" to read as the following: "The multi-output linear regression models have no hidden nodes and no nonlinearities, but are otherwise identical to the neural networks. These models therefore receive atmospheric state variables as inputs, which are then connected to eight output nodes, after which a softmax operator is applied to transform the output into a normalized likelihood. This method therefore does not allow nonlinearities, but does still permit the model to identify patterns unique to each phase of the MJO."

With that said, we finally address the reviewer's suggestion that the reduced accuracy in the multi-output linear regression approach is caused by low accuracies only for phases 1 and 8 of the MJO. The accuracies for each phase using the neural network and multi-output linear regression approaches averaged throughout the year are as follows:

Phase 1: Neural network: 47.5 Multi-output linear regression: 30.2 Phase 2: Neural network: 74.7 Multi-output linear regression: 40.3 Phase 3: Neural network: 76.1 Multi-output linear regression: 32.6 Phase 4: Neural network: 39.8

СЗ

Multi-output linear regression: 27.1 Phase 5: Neural network: 38.1 Multi-output linear regression: 25.3 Phase 6: Neural network: 79.1 Multi-output linear regression: 36.3 Phase 7: Neural network: 71.4Multi-output linear regression: 35.8 Phase 8:

Neural network: 56.4

Multi-output linear regression: 20.6

The accuracy of the multi-output linear regression approach is lower than that of the neural network for all phases, not just in phases 1 and 8. We also note that the minimum accuracies for both the multi-output linear regression and neural network approaches both occur during phases 4 and 5. The Maritime Continent has been shown to disrupt the spatial and temporal evolution of the MJO (e.g. Chen et al., 2020; Demott et al., 2018; Zhang and Ling, 2017), which likely makes it more difficult to identify its phase in spatial fields of atmospheric state variables during these phases. The reduced accuracy during phases 1 and 8 may be caused by these phases being associated with the initiation and demise of an MJO event, for which the atmospheric signature of the MJO may be weakest. These hypotheses extend beyond the scope of our paper, and we are interested to see if further studies use our proposed method to test such hypotheses more directly.

We again thank the reviewer for their quick response, and we hope our clarification and

proposed changes to wording address their comments.

References:

Chen, G., Ling, J., Li, C., Zhang, Y., Zhang, C. (2020). Barrier Effect of the Indo-Pacific Maritime Continent on MJO Propagation in Observations and CMIP5 Models. Journal of Climate, 33(12), 5173-5193.

DeMott, C. A., Wolding, B. O., Maloney, E. D., Randall, D. A. (2018). Atmospheric mechanisms for MJO decay over the Maritime Continent. Journal of Geophysical Research: Atmospheres, 123(10), 5188-5204.

Zhang, C., Ling, J. (2017). Barrier effect of the Indo-Pacific Maritime Continent on the MJO: Perspectives from tracking MJO precipitation. Journal of Climate, 30(9), 3439-3459.

Figure for Response Caption 1:

Schematic for the multi-output linear regression method used in this study. The first layer ingests vectorized input images and transfers this information to an output layer of 8 nodes that correspond to the eight phases of the MJO. A separate multi-output linear regression model is trained for each calendar week of the year

**Multi-Output Linear Regression for MJO Phase Identification**

Fig. 1.

---

## Referee Comment (RC2) · Anonymous Referee #2 · 4 Sep 2020

Recommendation:

Manuscript requires minor revisions

General comments:

The authors develop and apply neural network techniques to identify the phase, associated spatial structure and seasonality of the Madden–Julian Oscillation from a set of fundamental physical quantities (temperature, geopotential height, moisture and

winds). The authors aim to demonstrate that computational science techniques, such as these neural networks, can not only recover our current physical understanding of an important geophysical phenomenon like the MJO, but can also give us insights into the phenomenon to further that knowledge. This is an admirable, physically motivated application of computational techniques that are too often treated as "black boxes". The authors demonstrate that their neural network technique outperforms conventional linear regression; can identify the seasonality of the MJO, particularly with regard to the classical distinction between eastward propagation in boreal winter and northward propagation in boreal summer; and can aid understanding of which atmospheric variables are most "relevant" for the MJO in each phase and at each time of year.

This is a well-presented manuscript with clear objectives and sound understanding. I have a few minor comments for the authors to consider in a revised manuscript, detailed below, but otherwise I believe the manuscript should be accepted for publication after a round of minor revisions.

Minor revisions recommended:

1. For a discussion on the current understanding of MJO theory, and the historical evolution of that understanding, the authors may wish to refer to the recently published manuscript of Jiang et al. (2020): http://dx.doi.org/full/10.1029/2019JD030911.

2. Many in the target audience for this manuscript (climate scientists studying the MJO) will not be familiar with neural network techniques. Adding some background information, or references for further information, to section 2.2 would help the community to understand and accept these techniques. In particular, it would help to understand what "hidden layers", the "ReLu activation function" and the "softmax operator" are. These are probably commonly used terms in computational science, but I believe the authors would agree that they want to

avoid their audience treating this technique as a black box. A few sentences of explanation or a few references with further information would help guard against this.

3. In Figure 4, the authors show the probabilistic performance as a 2D histogram of predicted phase against target phase. A similar figure for the deterministic performance would be useful, to demonstrate whether the neural network technique performs similarly well for all target phases of the MJO. It is not easy to determine this from Fig. 4a, as the reader has to estimate the density of dots on the phase diagram.

4. Further to the above, from Fig. 4a it seems that the neural network performs better for stronger MJO events, as there seem to be more red dots closer to the unit circle and more blue and grey dots further away from the unit circle. Did the authors examine performance as a function of target MJO amplitude?

5. In Figure 5, the authors show the seasonality of the deterministic performance of the neural network technique, but provide little interpretation of the seasonality of performance. Can we learn anything – either about the MJO or about the neural network technique – from the fact that the neural networks are less successful at predicting MJO phase in boreal summer than in boreal winter? Can these results help to support the authors' conclusions about the seasonality of the MJO itself?

6. Related to the above, are there similar seasonalities in the probabilistic performance of the neural network technique? If so, is there any useful information we can gain from interpreting those seasonalities?

7. In Figure 7, the authors compare classical composite diagrams of OLR anomalies by MJO phase (panels (a) and (b)) against the "interpreted" results from the neural network that highlight the most salient features for identifying the MJO phase. The authors' interpretation is that the neural network identifies a more

focused area of active and suppressed convection as relevant for the MJO, versus the more widespread or diffuse anomalies in the classical composites. The common approach in composite analysis is to show only those anomalies that are statistically significant at some threshold (e.g., 5% significance) based on a $t$ test or similar. Did the authors perform such a test on panels (a) and (b)? If not, I would recommend performing one, as it might result in a more "focused" composite anomaly.

8. The results presented in this manuscript are certainly a useful first step toward using neural network techniques for understanding and predicting the MJO. However, the greatest uncertainty in community understanding of the MJO is not the identification of MJO phase or seasonality, but the mechanisms for MJO genesis, intensification and propagation. For instance, why do some MJO events propagate across the Maritime Continent while others do not? Why are some MJO events stronger than others? The authors hint that their neural network techniques might be useful for addressing these challenges (L315), but I believe a more detailed discussion of this potential would help the community to see the value in these techniques for understanding and predicting the MJO. As I am not an expert in neural network techniques, I cannot see a straightforward way to apply these techniques to understanding the propagation of the MJO or the mechanisms that drive that propagation. Can the authors add to this discussion in a revised manuscript?

9. Throughout section 3.2.2, the authors discuss the atmospheric fields that are most "relevant" to the MJO. Perhaps this word has a precise definition in neural network analysis, but I struggled with the interpretation here. What does "relevant" mean? Does it mean that the atmospheric field controls MJO strength, or determine MJO phase? Is a "relevant" field simply a field that has a structure common to most MJO events in that phase, regardless of intensity?

---

## Author Comment (AC2) · 21 Dec 2020

Authors' response to anonymous referee #2; GMD 2020-152

We appreciate the thoughtful comments from the second anonymous referee. Our responses to each comment are provided below.

Comment 1: For a discussion on the current understanding of MJO theory, and the historical evolution of that understanding, the authors may wish to refer to the recently published manuscript of Jiang et al. (2020): http://dx.doi.org/full/10.1029/ 2019JD030911.

Response: We have added this additional reference to the introduction.

[Figure]

Comment 2: Many in the target audience for this manuscript (climate scientists study-ing the MJO) will not be familiar with neural network techniques. Adding some back-ground information, or references for further information, to section 2.2 would help the community to understand and accept these techniques. In particular, it would help to understand what "hidden layers", the "ReLu activation function" and the "softmax op-erator" are. These are probably commonly used terms in computational science, but I believe the authors would agree that they want to avoid their audience treating this technique as a black box. A few sentences of explanation or a few references with further information would help guard against this.

Response: We have added a few citations to the end of Section 2.2 that can guide the reader to publications and books with more extensive details on the methodological details of neural networks. The focus of our paper is on a scientific application of neural networks, so we leave the reader to the extensive amount of free educational material available on the internet to learn more about neural networks.

Comment 3: In Figure 4, the authors show the probabilistic performance as a 2D his-togram of predicted phase against target phase. A similar figure for the deterministic performance would be useful, to demonstrate whether the neural network technique performs similarly well for all target phases of the MJO. It is not easy to determine this from Fig. 4a, as the reader has to estimate the density of dots on the phase diagram.

Response: We now list this information in the text in the first paragraph of Section 3.1.

Comment 4: Further to the above, from Fig. 4a it seems that the neural network performs better for stronger MJO events, as there seem to be more red dots closer to the unit circle and more blue and grey dots further away from the unit circle. Did the authors examine performance as a function of target MJO amplitude?

Response: We did not explicitly evaluate the accuracy of the neural network as a function of MJO amplitude. The review is correct in that the neural network is more accurate for higher amplitude cases, which is likely related to the MJO signal being

more prominent compared to non-MJO signals in these cases.

Comment 5: In Figure 5, the authors show the seasonality of the deterministic performance of the neural network technique, but provide little interpretation of the seasonality of performance. Can we learn anything – either about the MJO or about the neural network technique – from the fact that the neural networks are less successful at predicting MJO phase in boreal summer than in boreal winter? Can these results help to support the authors' conclusions about the seasonality of the MJO itself?

Response: The reduced accuracy during the summertime months is possibly related to the MJO comprising a smaller percentage of the total OLR variability during these months (e.g. Kiladis et al., 2014). Because there is more non-MJO signal, the MJO signal is muddled and therefore more difficult to identify. It is also possible that the MJO exhibits even more spatial nonlinearity during boreal summer, and that our chosen neural network architecture would therefore need more nonlinearity in order to identify the summer and winter modes with similar accuracy. This is an interesting topic for future study, and we have added a statement about this possibilities to the text of the last paragraph in Section 3.1.

Comment 6: Related to the above, are there similar seasonalities in the probabilistic performance of the neural network technique? If so, is there any useful information we can gain from interpreting those seasonalities?

Response: Yes, this is another good point. There are indeed seasonalities in the probabilistic performance. The probability distribution is more tightly clustered about the correct phase for boreal winter and more disperse for boreal summer. This is also reflected in the accuracies for boreal summer being lower than boreal winter. We don't think there is much meaningful insight to be had here except for the fact that the neural network is more uncertain and thus has lower accuracy during boreal summer. There may be interesting physical explanations for the greater uncertainty/reduced accuracy during boreal summer, although such an analysis would extend the scope of the paper

beyond its current core focus of proving base-line applicability of interpretable neural networks to geoscientific studies.

Comment 7: In Figure 7, the authors compare classical composite diagrams of OLR anomalies by MJO phase (panels (a) and (b)) against the "interpreted" results from the neural network that highlight the most salient features for identifying the MJO phase. The authors' interpretation is that the neural network identifies a more focused area of active and suppressed convection as relevant for the MJO, versus the more widespread or diffuse anomalies in the classical composites. The common approach in composite analysis is to show only those anomalies that are statistically significant at some threshold (e.g., 5% significance) based on a t test or similar. Did the authors perform such a test on panels (a) and (b)? If not, I would recommend performing one, as it might result in a more "focused" composite anomaly.

Response: Layerwise relevance propagation itself does not take into account significance, so we did not complete any significance testing on panels (a) and (b). A method for testing the significance of LRP heatmaps and optimal input fields is being developed separately, and will be usable in subsequent manuscripts. For this reason, we do not feel it is justified to filter the regression maps shown in subpanels (a) and (b) for significance. We agree that removing statistically insignificant regions from figures (a), (b), (c), and (d) may further limit the expanse of both the regression-based (panels a and b) and neural network-based (panels c and d) interpretations of the MJO.

Comment 8: The results presented in this manuscript are certainly a useful first step toward using neural network techniques for understanding and predicting the MJO. However, the greatest uncertainty in community understanding of the MJO is not the identification of MJO phase or seasonality, but the mechanisms for MJO genesis, intensification and propagation. For instance, why do some MJO events propagate across the Maritime Continent while others do not? Why are some MJO events stronger than others? The authors hint that their neural network techniques might be useful for addressing these challenges (L315), but I believe a more detailed discussion of this po-

tential would help the community to see the value in these techniques for understanding and predicting the MJO. As I am not an expert in neural network techniques, I cannot see a straightforward way to apply these techniques to understanding the propagation of the MJO or the mechanisms that drive that propagation. Can the authors add to this discussion in a revised manuscript?

Response: We have added a few lines of discussion on how a similar approach to the one used in this manuscript could be used for these specific hypotheses.

Comment 9: Throughout section 3.2.2, the authors discuss the atmospheric fields that are most "relevant" to the MJO. Perhaps this word has a precise definition in neural network analysis, but I struggled with the interpretation here. What does "relevant" mean? Does it mean that the atmospheric field controls MJO strength, or determine MJO phase? Is a "relevant" field simply a field that has a structure common to most MJO events in that phase, regardless of intensity?

Response: For clarity, we have changed the phrase "relevant" to "important for the identification of the MJO", or something similar to that for all cases. The phrase "relevant" does not have specific meaning in the computer science community, aside from the concept that LRP identified aspects of the input are most relevant to the network's associated output.